# Par3 interacts with Prickle3 to generate apical PCP complexes in the vertebrate neural plate

Ilya Chuykin[†], Olga Ossipova[†], Sergei Y Sokol*

Department of Cell, Developmental and Regenerative Biology, Icahn School of Medicine at Mount Sinai, New York, United States

**Abstract** Vertebrate neural tube formation depends on the coordinated orientation of cells in the tissue known as planar cell polarity (PCP). In the *Xenopus* neural plate, PCP is marked by the enrichment of the conserved proteins Prickle3 and Vangl2 at anterior cell boundaries. Here we show that the apical determinant Par3 is also planar polarized in the neuroepithelium, suggesting a role for Par3 in PCP. Consistent with this hypothesis, interference with Par3 activity inhibited asymmetric distribution of PCP junctional complexes and caused neural tube defects. Importantly, Par3 physically associated with Prickle3 and promoted its apical localization, whereas overexpression of a Prickle3-binding Par3 fragment disrupted PCP in the neural plate. We also adapted proximity biotinylation assay for use in *Xenopus* embryos and show that Par3 functions by enhancing the formation of the anterior apical PCP complex. These findings describe a mechanistic link between the apical localization of PCP components and morphogenetic movements underlying neurulation.

DOI: https://doi.org/10.7554/eLife.37881.001

## Introduction

Planar cell polarity (PCP) is a common phenomenon that refers to the orientation of cells in the plane of the tissue. PCP requires the functions of several conserved core proteins, Prickle, Van Gogh/Strabismus, Dishevelled, Frizzled and Flamingo/Stan, originally identified in *Drosophila* genetic studies. In *Drosophila* epithelial tissues, PCP is manifested by the distribution of the Frizzled/Dishevelled and Prickle/Van Gogh membrane complexes to opposite domains inside each cell (*Adler, 2012*; *McNeill, 2010*; *Peng and Axelrod, 2012*). In addition to planar polarity, vertebrate PCP proteins have been implicated in a variety of cell behaviors including cell migration, intercalation and apical constriction (*Gray et al., 2011*; *Ossipova et al., 2015b*; *Sokol, 1996*; *Sokol, 2015*; *Wallingford, 2012*; *Wallingford et al., 2000*). Disruption of PCP in vertebrates results in many embryonic defects including shortened body axes and opened neural tubes (*Ciruna et al., 2006*; *Sokol, 2000*; *Wallingford, 2012*; *Ybot-Gonzalez et al., 2007*). The existing models propose that PCP is established and maintained by mutually antagonistic interactions of core PCP complexes inside each cell and by positive feedback regulation between neighboring cells (*Adler, 2012*; *McNeill, 2010*). However, the molecular basis for the segregation of PCP complexes in polarized cells remains to be understood.

The outer cell layer of the vertebrate neural plate is an epithelium with clear apical-basal polarity (*Nikolopoulou et al., 2017*; *Nishimura et al., 2012*; *Suzuki et al., 2012*; *Wallingford et al., 2013*). The neuroepithelial cells also polarize along the anteroposterior embryonic axis with Prickle and Van Gogh-like 2 (Vangl2) proteins accumulating at the anterior *apical* cell corners (*Butler and Wallingford, 2018*; *Ossipova et al., 2015c*; *Sokol, 2015*). The apical accumulation of PCP components has been reported in other tissues, including the fly wing (*Axelrod, 2001*; *Bastock et al., 2003*;

**\*For correspondence:**
sergei.sokol@mssm.edu

[†]These authors contributed equally to this work

**Competing interests:** The authors declare that no competing interests exist.

*Wu et al., 2004*), the mouse node (*Antic et al., 2010*; *Mahaffey et al., 2013*; *Minegishi et al., 2017*) and zebrafish and frog neuroectoderm (*Ciruna et al., 2006*; *Ossipova et al., 2014*; *Ossipova et al., 2015c*). Currently, the significance of the apical accumulation of PCP proteins for tissue polarity is unknown. One possibility is that the formation of functional PCP complexes depends on their presence at the apical junctions, a cell compartment that is critically important for epithelial morphogenesis (*Takeichi, 2014*). This question can be addressed by studies of proteins regulating the formation of the apical domain and apical junctions.

The Par complex composed of Par6, Par3 and aPKC is among key regulators of the apical domain of the cell (*Joberty et al., 2000*; *Lin et al., 2000*; *Nance and Zallen, 2011*; *Suzuki and Ohno, 2006*). The conserved scaffold Par3/Pard3 plays a central role in this complex by interacting with multiple proteins and regulating cell polarity, adhesion, asymmetric cell division and migratory behavior in many tissues (*Afonso and Henrique, 2006*; *Bryant et al., 2010*; *Ebnet et al., 2001*; *Goldstein and Macara, 2007*; *Tawk et al., 2007*). Bazooka/Par3 and its associated proteins have been functionally linked to PCP in specific *Drosophila* tissues (*Beati et al., 2018*; *Blankenship et al., 2006*; *Djiane et al., 2005*; *Harris and Peifer, 2007*; *Simões et al., 2010*; *Wasserscheid et al., 2007*; *Zallen and Wieschaus, 2004*). Additionally, the effects of core PCP components on Par3 localization have been demonstrated in fly photoreceptor cells and sensor organ progenitors (*Aigouy and Le Bivic, 2016*; *Banerjee et al., 2017*; *Bellaïche et al., 2004*; *Besson et al., 2015*). In vertebrates, a recent study also suggested a link between Par3 and PCP (*Lin and Yue, 2018*), but whether Par3 itself is planar polarized, and how it modulates the activity of core PCP proteins has not been investigated.

To address this issue, we examined the localization and function of Par3 in the *Xenopus* neural plate. We report that Par3 is polarized in the plane of the neuroepithelium and functions in neural tube closure. Mechanistically, we find that Par3 associates with Prickle3 (Pk3) and recruits it to the apical cell membrane. Demonstrating the importance of this interaction, a specific Pk3-binding domain of Par3 interfered with the polarization of neuroepithelial cells. To further study PCP mechanisms, we developed an efficient in vivo proximity biotinylation approach using Pk3 fused to a bacterial biotin ligase. Using this assay, we demonstrate a novel role of Par3 in promoting the interaction of Pk3 and Vangl2 in neuroepithelial cells. These findings link the subcellular localization of two core PCP components to morphogenetic events underlying vertebrate neural tube closure.

## Results

### Planar polarization of Par3 in the *Xenopus* neural plate

Given the central role of Par3 in apical domain formation in many cell types (*Afonso and Henrique, 2006*; *Bryant et al., 2010*; *Joberty et al., 2000*), we examined its localization in the *Xenopus* neural plate, using characterized anti-Par3 antibodies (*Moore et al., 2013*; *Williams et al., 2014*). As reported for other cell types, Par3 was mainly localized at the apical membrane of neuroepithelial cells. Unexpectedly, e*n face* immunostaining of neural plate explants revealed the enrichment of Par3 at the cell junctions that were perpendicular to the anteroposterior body axis (*Figure 1A,B*). By contrast, ZO1 was homogeneously distributed to both anteroposterior and mediolateral cell junctions (*Figure 1C,D*). This polarized distribution of Par3 was verified by co-staining with ZO1 (*Figure 1E–E'', F*). Par3 antibody specificity was confirmed by lack of staining in the cells depleted of Par3 (*Figure 1G–G'*). The observed polarization of Par3 was similar to the enrichment of core PCP proteins and F-actin cables in the vertebrate neural plate (*McGreevy et al., 2015*; *Nishimura et al., 2012*; *Ossipova et al., 2015c*). Thus, our data suggest that Par3 may participate in PCP signaling in the neural plate.

### Par3 plays an essential role in neural plate PCP

To evaluate whether Par3 is required for neural plate PCP, we designed two Par3-specific morpholino oligonucleotides (MOs) with different sequences and confirmed their efficacy. (*Figure 1G* and *Figure 2—figure supplement 1*). Unilateral injection of these MOs but not the control MO into the prospective neuroectoderm inhibited neural tube closure in the majority of embryos (*Figure 2A–C* and *Figure 2—figure supplement 1B,C*), consistent with the known roles of PCP proteins in neural tube morphogenesis (*Wallingford et al., 2013*). Importantly, this defect was rescued by Myc-Par3

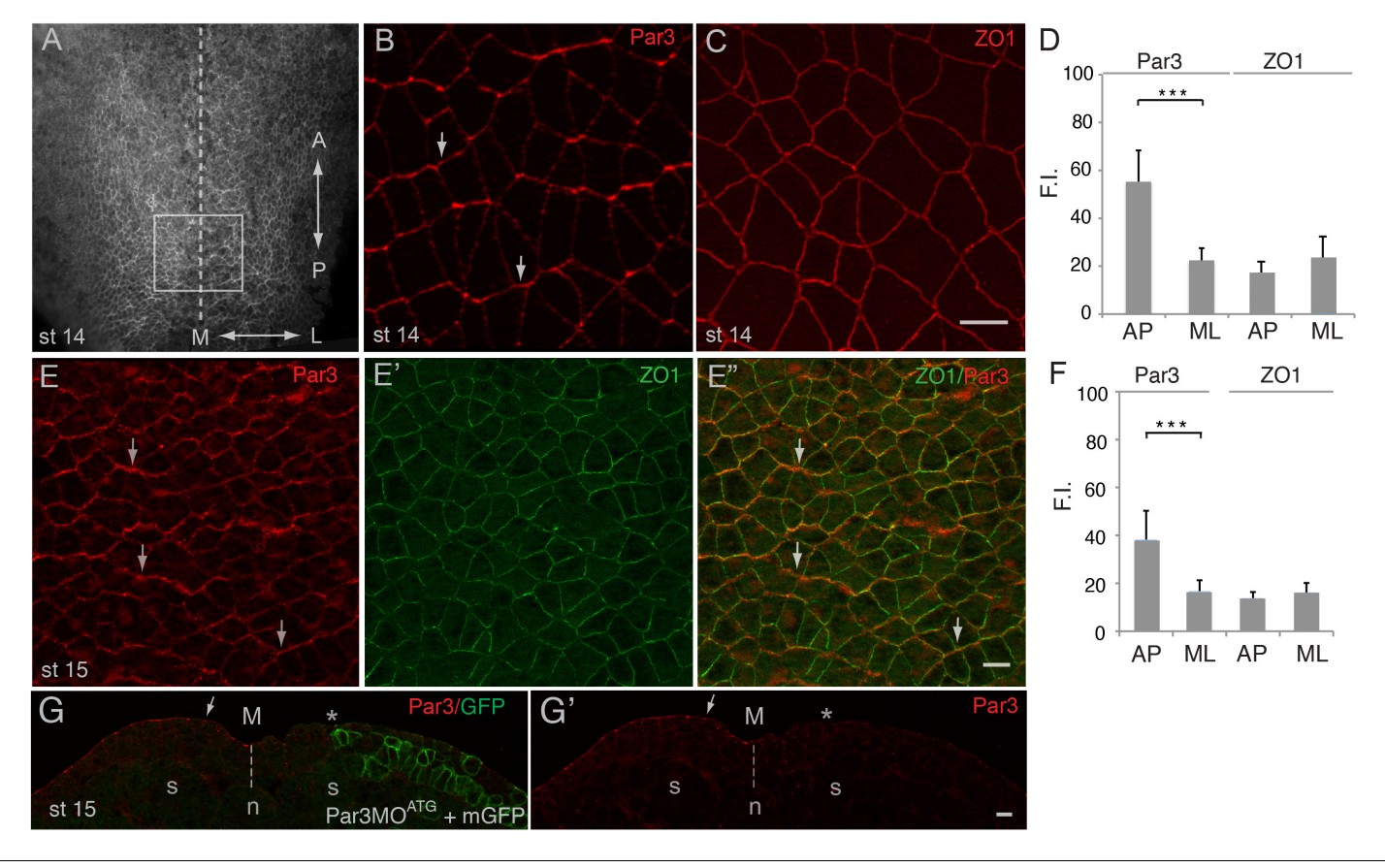

**Figure 1.** Planar polarization of Par3 in the *Xenopus* neural plate. (A–C, E) Representative images show *en face* view of immunostained neural plate explants prepared from fixed *Xenopus* embryos at stages (st) 14–15. (A) Neural plate explant with the approximate position of the imaged area (B–E) (boxed). Dashed line indicates the neural midline (M). The anteroposterior (AP) and the mediolateral (ML) axes are shown. (B) Par3 is enriched at AP, horizontal, cell borders (arrows) as compared to ML, vertical, cell borders. (C) ZO1 is equally distributed to all junctions. (D, F) Fluorescence intensity (F. I.). Means ± s. d. represent three different experiments. At least 30 cells from three to four different embryos were analyzed per group. Significance was determined by the two-tailed Student's t-test, p<0.001. (E–E") Double staining with Par3 and ZO1-specific antibodies reveals planar polarization of Par3 but not ZO1 along the AP axis. (G, G') Validation of the Par3 antibody. Cross-section of a neurula embryo, stage 15, unilaterally injected with Par3MO$^{ATG}$ (20 ng, asterisk) and GFP-CAAX, membrane GFP (mGFP) RNA (100 pg) as lineage tracer. Arrow points to apical Par3 at the uninjected side, whereas the neural plate is flat on the injected side; n, notochord; s, somite, M, midline (dashed line). Control MO did not alter Par3 distribution (see *Figure 2—figure supplement 3*). Scale bars are 20 μm.

DOI: https://doi.org/10.7554/eLife.37881.002

RNA indicating specificity (*Figure 2C–C'*). We next analyzed the localization of Vangl2 in the morphants. Vangl2 polarization at the anterior apical cell borders was disrupted in Par3-deficient neural plate cells, but not in wild-type cells or those injected with control MO (*Figure 2D–F*). Cryosections revealed that Vangl2 is enriched basolaterally rather than apically after Par3 depletion (*Figure 2— figure supplement 2*). The total amount of Vangl2 protein was not altered (*Figure 2G*), indicating effect on anterior cortical localization rather than protein stability. Notably, in these experiments, ectoderm-targeted MO injections did not significantly affect apicobasal polarity and junctional markers, including aPKC, ZO1, and β-catenin (*Figure 2—figure supplement 3*). We also found that planar polarization of Par3 was reduced in the cells depleted of Vangl2 (*Figure 2—figure supplement 4*), suggesting that Par3 and Vangl2 reinforce each other localization in the neuroepithelium. These observations suggest a novel function for Par3 in vertebrate PCP and reiterate the existence of regulatory feedback between Par3 and the core PCP machinery.

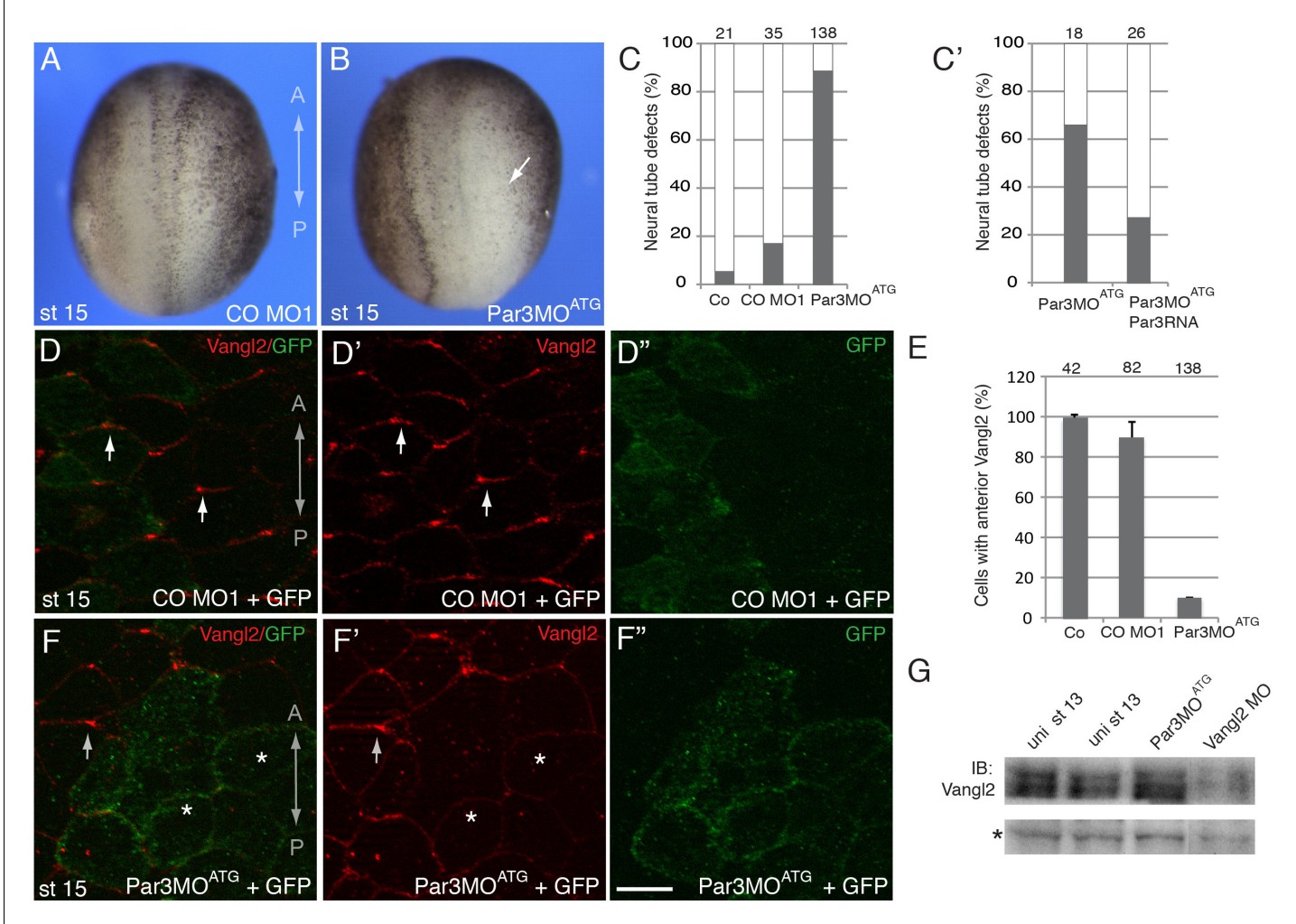

**Figure 2.** A requirement for Par3 in neural plate PCP and neural tube closure. (A–F), Eight-cell embryos were unilaterally injected into two animal blastomeres with control morpholino 1 (CO MO1), 20 ng, or Par3MO^ATG, 20 ng, as indicated, with GFP RNA, 0.1 ng, as a lineage tracer. Dorsal view is shown, and the anteroposterior (AP) axis of the neural plate and embryonic stage 15 (st 15) are indicated. (A–C) Par3 depletion results in neural tube defects. Arrow points to the open neural fold. (C) Frequencies of neural tube defects were scored by the lack of neural fold formation. Numbers of scored embryos per group are shown above each bar. (C') Partial rescue of the defect with Par3 RNA, 0.2 ng, is shown. Data are from three different experiments. (D–F), Embryos were injected as described above. Neural plate cells mosaically depleted of Par3 (labeled by GFP) lack Vangl2 enrichment at the anterior border of each cell (asterisks) as compared to control GFP-negative cells (arrows). D', D'', F', F'' are single-channel images corresponding to D and F. CO MO1 injection had no effect on the anterior distribution of Vangl2. Scale bar, 20 μm. (E) Quantification of data from the experiments with Par3MO^ATG showing mean frequencies ± s. d. of cells with anterior Vangl2. At least 5–10 embryos were examined per each treatment. Numbers of scored cells are shown on top of each bar. Co, uninjected control. (G) Immunoblot analysis of Vangl2 in embryo extracts. *Xenopus* embryos were injected with the indicated MOs into animal pole blastomeres at the two-cell stage and collected at stage 13 for immunoblotting (IB) with Vangl2 antibodies. Asterisk marks a non-specific band indicating loading. Uni, uninjected.

DOI: https://doi.org/10.7554/eLife.37881.003

The following figure supplements are available for figure 2:

**Figure supplement 1.** Depletion of Par3 with Par3MO^5'UTR causes neural tube closure defects.

DOI: https://doi.org/10.7554/eLife.37881.004

**Figure supplement 2.** Par3MO^ATG inhibits apical accumulation of Vangl2 at the neural plate midline.

DOI: https://doi.org/10.7554/eLife.37881.005

**Figure supplement 3.** Par3 depletion does not affect the localization of aPKC, b-catenin and ZO1 in gastrula ectoderm.

DOI: https://doi.org/10.7554/eLife.37881.006

**Figure supplement 4.** Planar polarity of Par3 is lost in Vangl2-depleted cells in the neural plate.

DOI: https://doi.org/10.7554/eLife.37881.007

## The interaction of Par3 and Pk3

Given the involvement of Par3 in neural plate PCP, we examined whether Par3 forms a physical complex with core PCP proteins. We found that Par3 formed a complex with Pk3 (*Figure 3A*), a core PCP protein that is expressed in the superficial ectoderm (*Chu et al., 2016*; *Ossipova et al., 2015a*). Even stronger interaction was observed with Pk3ΔPET, a deletion mutant of Pk3 that lacks a conserved PET (Prickle, Espinas and Testin) domain (*Figure 3B,C*), consistent with a proposed role of the PET domain in intramolecular interactions that keep Prickle in an inactive conformation (*Sweede et al., 2008*). These immunoprecipitation experiments demonstrate the physical association of Par3 and Pk3.

Despite the encouraging results in HEK293T cells, our initial experiments did not detect the physical association of ectopically expressed Par3 and Pk3 in *Xenopus* embryos using conventional pull-downs (data not shown). We then decided to examine this interaction using proximity biotinylation, a highly sensitive approach (*Choi-Rhee et al., 2004*; *Roux et al., 2012*). We constructed a fusion of Pk3 with a promiscuous biotin ligase (BL) from *Aquifex aeolicus* (*Kim et al., 2016*). In the presence of biotin, this fusion is expected to biotinylate proteins in the immediate proximity of Pk3 in cells under physiological conditions (*Figure 3D*). To our knowledge, this assay has not been used in vivo in organisms developing at lower temperatures due to the concerns that BL would not be sufficiently active at 13–22°C as compared to 37°C. We generated BL fusions with Pk3 and Vangl2 and supplied them to early *Xenopus* embryos by microinjection of mRNAs together with biotin. We first compared different BL fusions and found that the conserved N-terminal portion is sufficient for promiscuous enzymatic activity as compared to the three domains present in BirA*, the original mutated biotin ligase from *E. coli* (*Choi-Rhee et al., 2004*; *Roux et al., 2012*). Robust autobiotinylation of the fusion protein has been detected as early as stages 11–12 both at 13°C and 24°C (data not shown). We next assessed whether Par3 is biotinylated by BL-Pk3 that would reflect an in vivo interaction and confirmed it in pulldown assays with anti-Par3 antibodies (*Figure 3E*). In support of assay specificity, no biotinylation of Par3 was detected in the presence of BL-Vangl2, another BL fusion protein. The enzymatic activity of BL-Vangl2 was verified by efficient biotinylation of Pk3 (data not shown). The biotinylation of both exogenous and endogenous Par3 was detected (*Figure 3E*). Together, these experiments indicate that Pk3 interacts with Par3 both in cultured cells in vitro and in frog embryos in vivo.

## The association of Par3 with the Pk3/Vangl2 complex and the identification of Pk3-interacting domains

We next assessed whether Par3 associates with Vangl2. Whereas Par3, on its own, did not bind Vangl2 in transfected HEK293T cells, we found that Vangl2 was efficiently pulled down with Par3 when Pk3 or Pk3ΔPET were co-expressed (*Figure 4A*). Notably, Vangl2 was not detected in pulldowns with Par3ΔΔ, a Par3 deletion mutant that does not bind Pk3 (see below), indicating that Pk3 binding is necessary for the Par3-Vangl2 interaction (*Figure 4A*). The simplest interpretation of this result is the formation of the ternary complex of Par3, Prickle3 and Vangl2. Notably, overexpressed Par3 enhanced the association of Pk3 and Vangl2 within the ternary complex.

Par3 consists of several domains including the N-terminal oligomerization domain, three PDZ domains, the aPKC-binding site and the carboxy-terminal region essential for cortical localization (*Krahn et al., 2010*; *Simões et al., 2010*). To identify the domains interacting with Pk3, several Par3 constructs were tested for their association with Pk3 in pull down experiments (*Figure 4B*). The C-terminus of Par3 and the fragment including the first two PDZ domains were sufficient for Pk3 binding, whereas the N-terminus, the PDZ3 domain and the aPKC-binding site did not show any binding (*Figure 4B,C*). Mutated Par3 proteins with single domain deletions were still able to interact with Pk3 (data not shown). However, the double deletion mutant lacking PDZ1/2 and the C-terminal domain (Par3ΔΔ) did not bind Pk3 and Pk3ΔPET (*Figure 4D*), demonstrating that these two regions of Par3 are involved in Pk3 association. Moreover, we found that wild type Par3, but not Par3ΔΔ, was biotinylated by BL-Pk3 in *Xenopus* embryos (*Figure 4E*), confirming the importance of the identified Pk3-binding domains for this interaction in vivo.

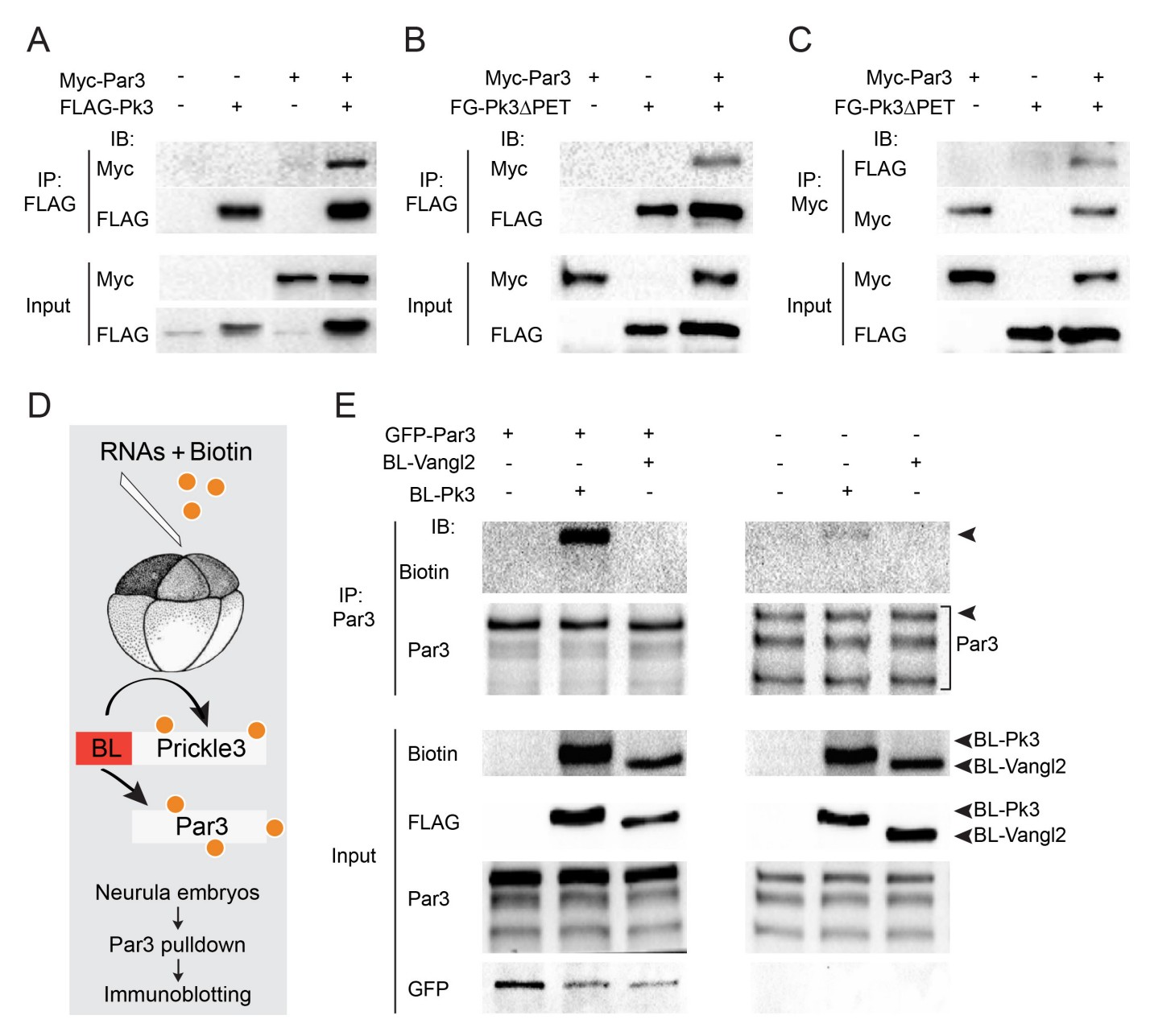

**Figure 3.** Par3 interacts with Pk3 in HEK293T cells and *Xenopus* embryos. (A–C) Physical interaction of Par3 and Pk3 in transfected HEK293T cells. Myc-Par3 is pulled down from cell lysates with FLAG-Pk3 (A) or FLAG-GFP(FG)-Pk3ΔPET (B). FG-Pk3ΔPET is pulled down with Myc-Par3 (C). (D, E) Interaction of Par3 and Pk3 in *Xenopus* embryos assessed by proximity biotinylation. (D) Experimental scheme. Biotin and RNAs encoding FLAG-Biotin Ligase (BL)-Pk3 or FLAG-BL-Vangl2, 0.5 ng each, with or without GFP-Par3 RNA, 0.1 ng, were injected into the animal region of four- to eight- cell embryos. Injected embryos were lysed at stages 12.5–13 for immunodetection of biotinylated proteins. (E) Exogenous (left) and endogenous (right) Par3 is biotinylated by BL-Pk3 but not BL-Vangl2. Three bands that correspond to endogenous Par3 isoforms (bracket) are pulled down and detected by anti-Par3 antibodies, however only the top band corresponds to exogenous GFP-Par3 (arrowheads). Protein levels are shown by immunoblotting with anti-Myc and anti-FLAG antibodies in (A–C) and anti-biotin, anti-Par3, anti-FLAG and anti-GFP antibodies in (E).
DOI: https://doi.org/10.7554/eLife.37881.008

## Functional significance of the Par3-Pk3 interaction for PCP

If the physical interaction of Pk3 and Par3 is critical for neuroepithelial polarization, a Pk3-interacting domain of Par3 would be predicted to interfere with PCP. We found that expression of Par3[272-544] but not Par3[545-756] construct in *Xenopus* embryos inhibited the interaction of Par3 and Pk3

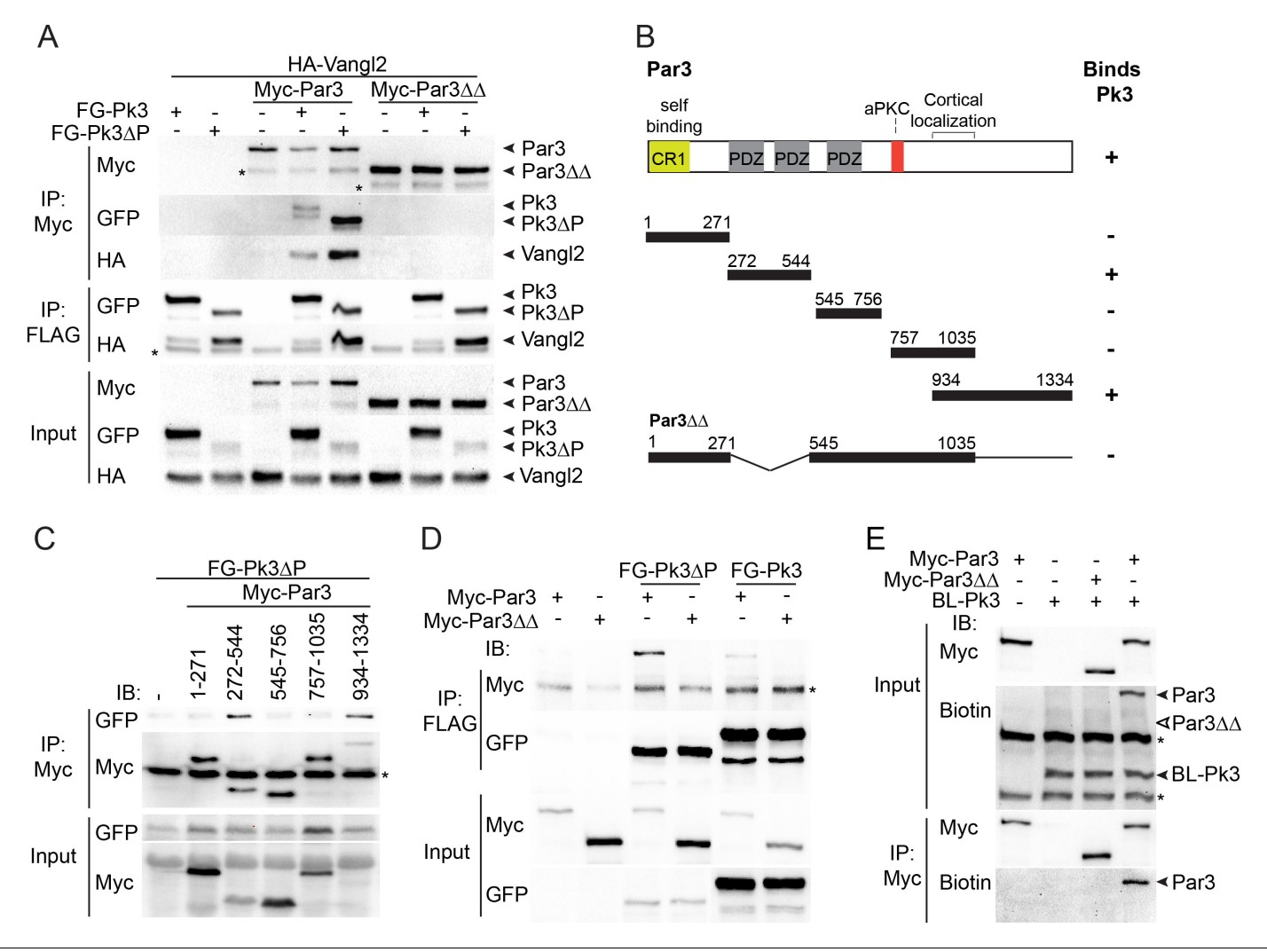

**Figure 4.** The association of Par3 with the Pk3/Vangl2 complex and the identification of Pk3-interacting domains. (**A**) HEK293T cells were transfected with Par3, Pk3 or Pk3ΔPET (Pk3ΔP) and Vangl2 constructs as indicated. Sequential pulldowns of protein complexes containing Myc-Par3, FLAG-GFP (FG)-Pk3 or FG-Pk3ΔP and HA-Vangl2 with Myc-Trap and anti-FLAG (M2) beads are shown. Note that Par3 binds Vangl2 only in the presence of Pk3 or Pk3ΔP. Asterisks mark nonspecific bands. (**B**). Schematic showing the Par3 constructs used in these experiments and the summary of Par3 binding. (**C**) Co-immunoprecipitation of Myc-Par3 constructs with FG-Pk3ΔP (see *Figure 3* legend). (A sterisk indicates IgG heavy chain. (**D**) Pulldowns of FG-Pk3ΔP or wild type FG-Pk3 with Myc-Par3 or Myc-Par3ΔΔ. Asterisk shows a nonspecific band. (**E**) Interaction of Par3 and Pk3 assessed by proximity biotinylation in *Xenopus* embryos. Exogenous Par3 but not Par3ΔΔ is biotinylated by BL-Pk3. Black arrowheads point to biotinylated Par3 and BL-Pk3, and white arrowheads indicate the expected position of Par3ΔΔ. Asterisks indicate endogenous proteins detected by anti-biotin antibodies. Protein levels are shown by immunoblotting with anti-Myc, anti-HA, anti-GFP and anti-biotin antibodies as indicated.
DOI: https://doi.org/10.7554/eLife.37881.009

as assessed by proximity biotinylation (*Figure 5—figure supplement 1*). Of note, Par3[272-544] did not affect Par3 apical localization (*Figure 5—figure supplement 2*). Next, we assessed whether the Pk3-binding Par3[272-544] fragment would influence the polarity of Pk3/Vangl2 complexes in the neural plate. We observed that Par3[272-544], but not the control fragment Par3[545-756], interfered with Pk3/Vangl2 complex polarization at the anterior cell borders (*Figure 5A–E*). These results are consistent with our hypothesis that the interaction of Par3 and Pk3 is essential for the formation of the functional Pk3/Vangl2 complex in neuroepithelial cells.

PCP signaling is commonly associated with gastrulation movements and body axis extension (*Gray et al., 2011*; *Ossipova et al., 2015b*; *Sokol, 2000*). Since we found that PDZ1/2 and the C-terminus of Par3 are required for the interaction between Par3 and Pk3 (*Figure 4*), we assessed its

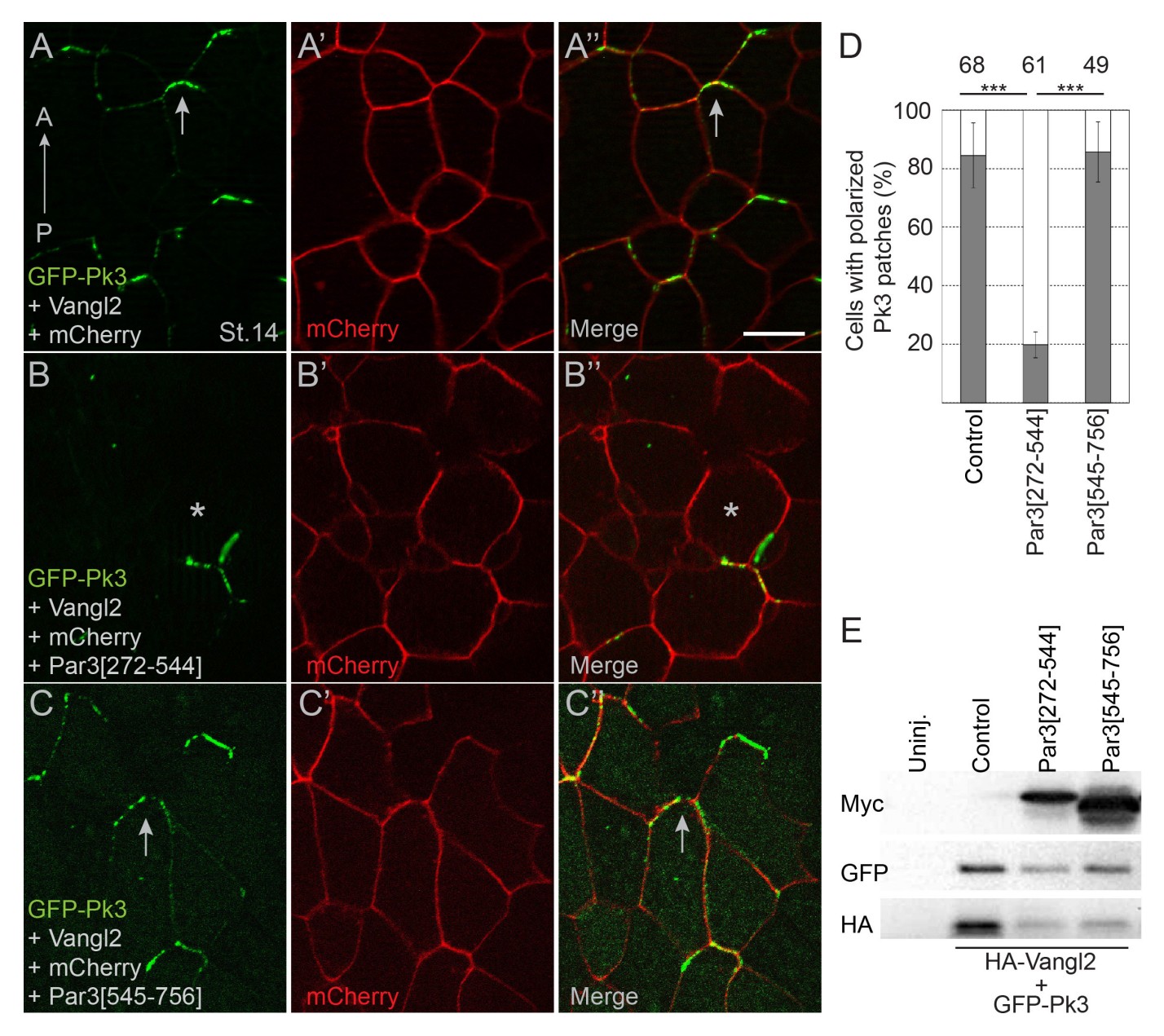

**Figure 5.** Pk3-interacting fragment of Par3 interferes with neural plate PCP. Two dorsal blastomeres of 16 cell embryos were injected with RNAs encoding GFP-Pk3 (100 pg), HA-Vangl2 (25 pg) and mCherry (70 pg) without (A–A'') or with Par3[272-544] (0.5 ng) (B–B'') or Par3[545-756] (0.5 ng) (C–C''). Cells from embryos at stage 14 (St.14) with anteriorly polarized (arrows) and mislocalized (asterisks) GFP-Pk3 patches are shown. Anteroposterior (AP) axis of the neural plate is indicated. Scale bar, 20 µm. (D) Quantification of data in (A–C) shown as mean frequencies ± s. d. of polarized GFP-Pk3 patches in neuroepithelial cells. Total numbers of scored cells are shown above each bar; 5 to 25 cells were scored per embryo with five embryos taken for each experimental condition, statistical significance was determined by two-tailed Student's t-test, p<0.001. Data are representative of two experiments. (E) Protein expression levels were assessed in stage 14 embryos by immunoblotting with anti-Myc, anti-GFP and anti-HA antibodies. Control, embryos injected with HA-Vangl2 and GFP-Pk3 RNAs without Par3 constructs, Uninj., uninjected embryos.

DOI: https://doi.org/10.7554/eLife.37881.010

The following figure supplements are available for figure 5:

**Figure supplement 1.** Interaction of Pk3 and Par3 in *Xenopus* embryos is inhibited by Pk3 binding fragment of Par3.

DOI: https://doi.org/10.7554/eLife.37881.011

**Figure supplement 2.** Par3[272-544] construct does not affect Par3 localization.

DOI: https://doi.org/10.7554/eLife.37881.012

**Figure supplement 3.** The ability of Par3 to inhibit blastopore closure and body axis elongation is lost upon the disruption of Pk3 binding.

*Figure 5 continued on next page*

Figure 5 continued

DOI: https://doi.org/10.7554/eLife.37881.013

functional relevance by comparing the activities of wild-type Par3 and the Par3 deletion mutant (Par3ΔΔ) that does not bind Pk3. Upon overexpression in prospective mesoderm, wild-type Par3 interfered with blastopore closure and body axis extension (as measured by embryo length), whereas Par3ΔΔ lacked these activities (*Figure 5—figure supplement 3A–F*). Both proteins were expressed at similar levels arguing that this lack of activity in Par3ΔΔ is not due to protein degradation (*Figure 5—figure supplement 3G*). This observation supports the view that the interaction of Par3 and Pk3 is critical for the function of Par3 in convergent extension movements of mesoderm and neural tube closure.

## Par3 recruits Pk3 to the apical surface

The interaction of Par3 and Pk3 suggests that Par3 functions in PCP by promoting the apical recruitment of PCP complexes in the neural plate. To test this possibility, we assessed the effect of Par3 on the subcellular distribution of Pk3 in gastrula ectoderm, the tissue that gives rise to the neural plate. Both endogenous and exogenous Par3 localized to the apical side of superficial ectoderm cells (*Figure 6A–C'*), consistent with other reports (*Afonso and Henrique, 2006*; *Grego-Bessa et al., 2016*; *Krahn et al., 2010*). Whereas Pk3 by itself was largely cytoplasmic, it was recruited to the apical cortex by Par3 (*Figure 6D–E''* and *Figure 6—figure supplement 1A–D*). By contrast, the punctate cytoplasmic distribution of GFP-Dvl2 did not change in the presence of Par3 (*Figure 6—figure supplement 1E,F*). Also, Pk3 localization was not altered by the cortically localized Myc-Par1/MARK (*Ossipova et al., 2007*) (*Figure 6—figure supplement 1G,H*), further confirming specificity. This apical recruitment of Pk3 by Par3 was verified with two differently tagged Pk3 constructs (*Figure 6—figure supplement 1A–D*).

These experiments suggest a novel molecular mechanism, in which Par3 recruits Pk3 to position the Pk3/Vangl2 complex to the apical surface.

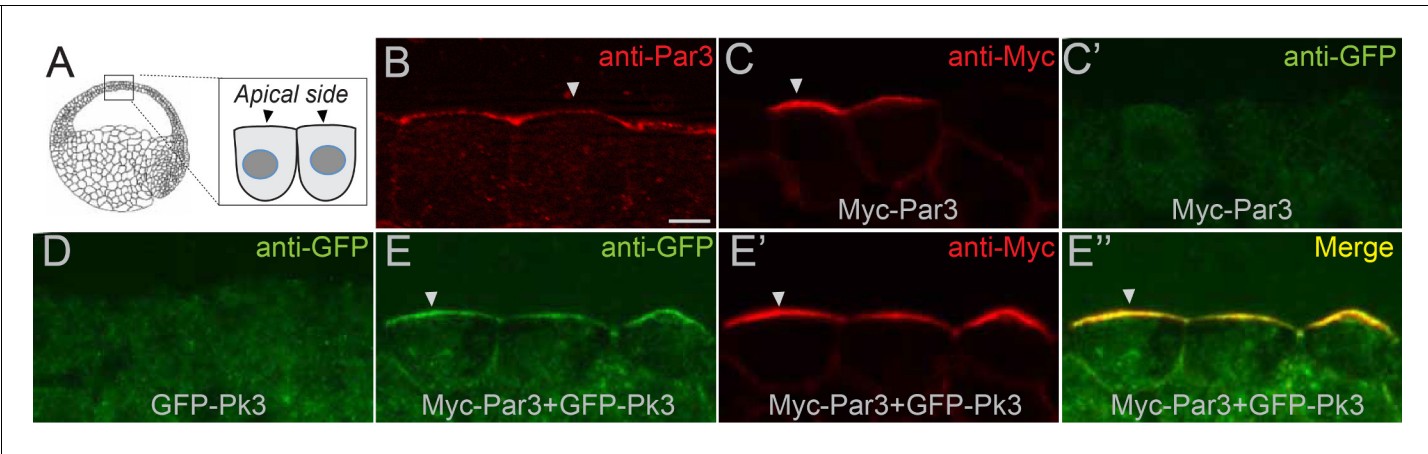

**Figure 6.** Par3 recruits Pk3 to the apical side of the cell in vivo. Embryos were injected with GFP-Pk3 and Myc-Par3 RNAs (100 pg each), cryosectioned at stage 10.5 and immunostained with indicated antibodies (A) Scheme showing a relative position of imaged superficial ectoderm cells. Both endogenous Par3 (B), and exogenous Myc-Par3 (C, E', E'') are apically localized. (C') Lack of Myc-Par3 staining with anti-GFP antibody. (D) Lack of apical enrichment of exogenous GFP-Pk3. (E–E'') Myc-Par3 recruits GFP-Pk3 to the apical surface. (B, C, E) Apical enrichment is shown by arrowheads. The apical recruitment of Pk3 was observed in > 90% of cells coinjected with Pk3 and Par3. Each group contained five embryos. The same results were obtained in five independent experiments. Scale bar 10 μm.

DOI: https://doi.org/10.7554/eLife.37881.014

The following figure supplement is available for figure 6:

**Figure supplement 1.** Par3 recruits Pk3 but not Dvl2 to the apical cell membrane.

DOI: https://doi.org/10.7554/eLife.37881.015

## Par3 is required for the formation of the apical PCP complex in the neural plate

We next studied how Par3 deficiency would affect the interaction of Pk3 and Vangl2 using proximity biotinylation assay. To interfere with the function of Par3 we microinjected Par3MO[5'UTR] or the N-terminal dimerization domain of Par3 (Par3N) that disrupts Par3 localization and function in *Xenopus* embryos and mammalian cultured cells (*Figure 7—figure supplement 1*)(*Mizuno et al., 2003*; *Werner et al., 2014*). We observed that Vangl2 was efficiently biotinylated by BL-Pk3 (*Figure 7A,B*), consistent with the previously reported interactions of Vang and Prickle and the colocalization of Vangl2 and Pk3 in the *Xenopus* neural plate and the epidermis (*Bastock et al., 2003*; *Chu et al., 2016*; *Chu and Sokol, 2016*; *Jenny et al., 2003*). Par3N strongly reduced the Vangl2 biotinylation in pulldown experiments (*Figure 7A*). Similarly, Vangl2 was less biotinylated in embryos depleted of Par3 (*Figure 7B*).

To further support the requirement of Par3 in PCP, we examined the polarization of the Pk3/Vangl2 complexes in neuroepithelial cells expressing Par3N. Consistent with our previous observations (*Chu and Sokol, 2016*; *Ossipova et al., 2015c*), exogenous Pk3 and Vangl2 proteins formed anterior cortical patches in the neural plate (*Figure 7C,D*). By contrast, after coinjection of Par3N RNA, these patches were randomly distributed around the cell cortex and lacked the anterior bias (*Figure 7E,F*). Similarly, Pk3/Vangl2 complexes were not polarized in cells depleted of Par3 (*Figure 7—figure supplement 2*). Moreover, unilateral injection of Par3N inhibited neural tube closure (*Figure 7G–I*). Together, these experiments support the essential role of Par3 in the polarization of core PCP proteins in the neural plate.

Overall, these data suggest that Par3 is required for the formation of the polarized Pk3/Vangl2 complex in the neural plate.

## Discussion

Our experiments demonstrate that Par3 polarizes in the plane of the *Xenopus* neural plate and is required for PCP and neural tube closure. Par3 is enriched at anteroposterior cell boundaries that are parallel to the mediolateral axis of the *Xenopus* neural plate, providing evidence for the polarization of a vertebrate apicobasal polarity protein in the plane of the tissue. Whereas the mechanism responsible for this planar polarization is not known, the same biochemical or mechanical signals that affect core PCP proteins (*Chien et al., 2015*; *Chu and Sokol, 2016*) (*Kim et al., 2018*) are likely to regulate Par3. One possibility is that the association of core PCP protein complexes with Par3 may directly contribute to Par3 enrichment at specific locations (*Banerjee et al., 2017*; *Besson et al., 2015*). In contrast to *Xenopus* Par3, fly Bazooka/Par3 is enriched at dorsoventral but not anteroposterior boundaries of intercalating cells during germ-band extension along the anteroposterior axis (*Simões et al., 2010*). Notably, in the germband, Bazooka polarity depends on Rho-associated kinase that shows complementary localization at anteroposterior cell borders (*Simões et al., 2010*). In the same tissue, Bazooka localization also requires the LIM-domain protein Smallish (*Beati et al., 2018*), but does not seem to depend on core PCP proteins (*Simões et al., 2010*; *Zallen and Wieschaus, 2004*). Nevertheless, in the fly eye epithelium Bazooka is planar polarized under the control of the core PCP protein Stan/Flamingo (*Aigouy and Le Bivic, 2016*). These studies indicate that tissue-specific mechanisms control Par3 planar polarization.

The distribution of Par3 in the neural plate is similar to the localization of other vertebrate PCP-related complexes (*Ciruna et al., 2006*; *Devenport and Fuchs, 2008*; *McGreevy et al., 2015*; *Nishimura et al., 2012*; *Ossipova et al., 2015c*; *Shindo and Wallingford, 2014*), suggesting a new role for Par3 in PCP. This role is supported by our loss-of-function experiments, in which the polarization of endogenous Vangl2 and exogenous Pk3/Vangl2 complexes were disrupted by Par3 depletion and the expression of the dominant interfering Par3N construct. These embryos developed neural tube abnormalities, consistent with defective PCP signaling (*Wallingford et al., 2013*). Importantly, despite the clear morphological and molecular manifestation of Par3 depletion at neurula stages, we detected no significant changes in the apicobasal polarity markers aPKC, ZO1, and β-catenin. Similarly, no apicobasal polarity defects have been reported for *par3* mutant follicular epithelial cells in *Drosophila* (*Shahab et al., 2015*). Also, apicobasal polarity markers did not change in retinal endothelial cells of *par3* knockout mice (*Hikita et al., 2018*). Thus, the requirement of Par3 for apicobasal polarity may depend on developmental context and cell type. Overall, our results suggest a

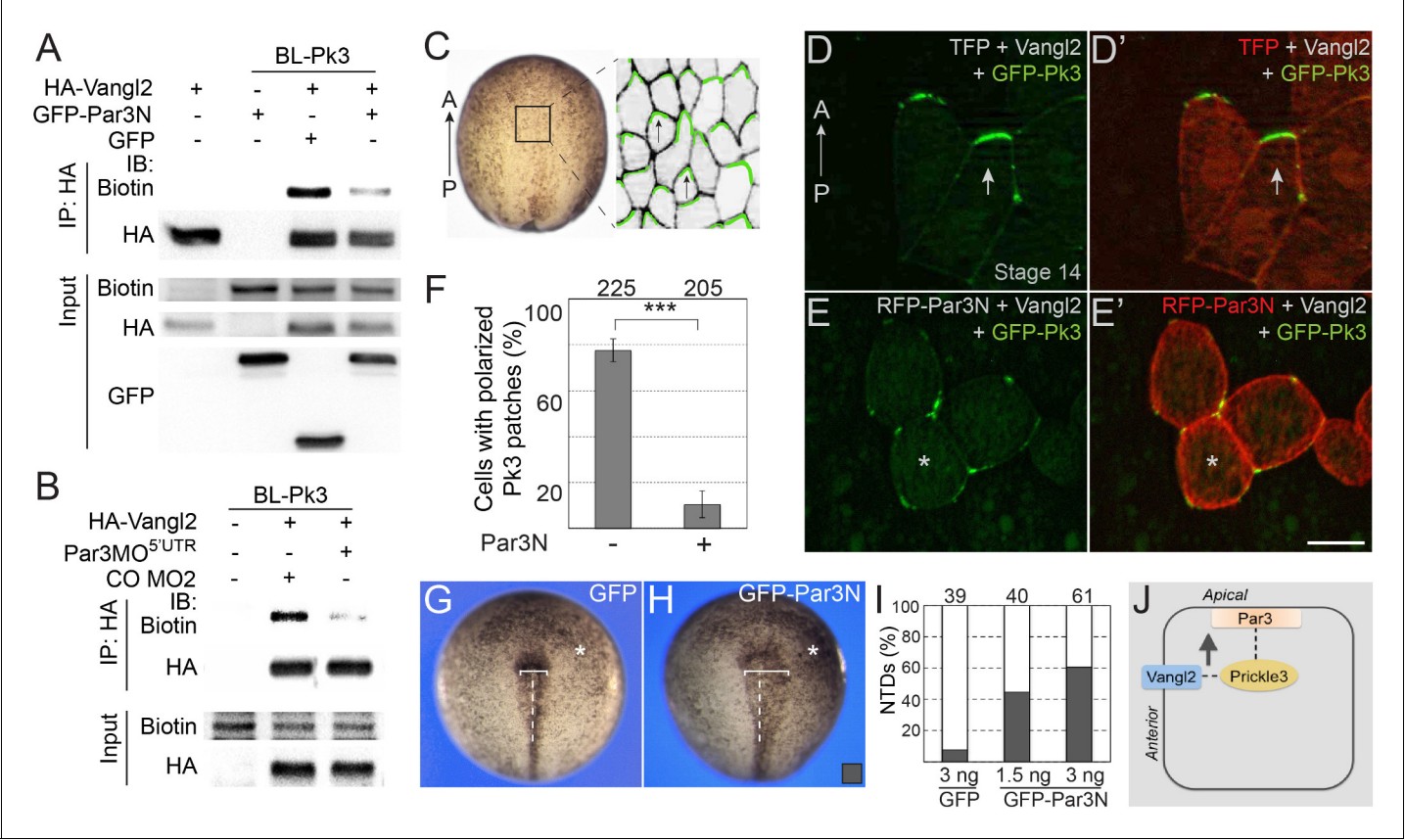

**Figure 7.** Par3 is required for the interaction of Pk3 and Vangl2 in vivo. (A, B) Decrease of Vangl2 biotinylation by BL-Pk3 (see *Figure 3*) in embryos injected with Par3MO or Par3N RNA. Embryos were coinjected into animal blastomeres with biotin and RNAs encoding BL-Pk3, 0.1 ng, and HA-Vangl2, 50 pg, GFP, 1 ng, GFP-Par3N, 1 ng (A), or MOs (CO MO2 or Par3MO[5'UTR], 10 ng each, (B) as indicated. Biotinylated HA-Vangl2 was detected with anti-biotin antibodies in pulldowns with anti-HA antibodies. (A, B). Protein expression levels in stage 13 embryos were assessed by immunoblotting of lysates with anti-HA, anti-GFP and anti-biotin (for FLAG-BL-Pk3 protein) antibodies (A, B). (C–I) Par3N disrupts neuroepithelial PCP. (C–E) Two dorsal blastomeres of 16 cell embryos were injected with GFP-Pk3 RNA, 100 pg, HA-Vangl2 RNA, 25 pg, and Turbo FP635 (TFP) RNA, 0.1 ng, or RFP-Par3N RNA, 0.4 ng. Injected embryos were cultured until stage 14, fixed and the neural plate explants were imaged. (C) *En face* view of a neurula embryo. Polarized cells from the boxed area used for imaging are shown schematically on the right. Anterior PCP complexes are in green (arrows), the anterior-posterior (AP) axis is indicated. (D–E') Representative images. (D, D') Anterior enrichment of Pk3/Vangl2 complexes (arrows) in a control embryo. (E, E') Lack of PCP (asterisks) in a Par3N-expressing embryo. Scale bar, 20 μm. (F) Frequencies of neuroepithelial cells containing polarized Pk3 cortical patches. Means ± s. d. are shown for three independent experiments. Total numbers of scored cells are above each bar. Significance was determined by the two-tailed Student's t-test, p<0.001 (asterisks). (G–I) Neural tube defects in representative stage 17/18 embryos unilaterally injected with GFP RNA, 3 ng, (G) or GFP-Par3N RNA, 3 ng, (H). Asterisk indicates the injected side. Note the difference in the distance between the neural fold and the midline (white line) at the injected side as compared to the uninjected side. (I) Frequencies of neural tube defects shown in G, H. Numbers of embryos from two experiments are shown above each bar. (J) Working model: Par3 recruits Pk3 to the apical surface to promote the interaction of Pk3 and Vangl2 at the apical junctions that is necessary for planar polarization.

DOI: https://doi.org/10.7554/eLife.37881.016

The following figure supplements are available for figure 7:

**Figure supplement 1.** Par3N inhibits the apical localization of Par3.
DOI: https://doi.org/10.7554/eLife.37881.017

**Figure supplement 2.** Par3 depletion disrupts the polarization of Pk3-Vangl2 complexes in the neural plate.
DOI: https://doi.org/10.7554/eLife.37881.018

direct role of Par3 in PCP, rather than a secondary effect on neural tube closure through the modulation of apicobasal polarity. The novel function of Par3 in the neural plate PCP is consistent with the requirement of Par3 in zebrafish neural tube morphogenesis (*Tawk et al., 2007*). Supporting our hypothesis, a recent study has implicated rare Par3 variants in human cranial neural tube defects (*Chen et al., 2017*). This contrasts lack of neural tube abnormalities in Par3 -/- mouse embryos

(*Hirose et al., 2006*) that could be due to functional redundancy (*Ishiuchi and Takeichi, 2011*; *Kohjima et al., 2002*).

Our experiments provide a mechanistic insight into how apicobasal polarity is linked to PCP. A proximity biotinylation approach that we adapted to use in *Xenopus* embryos demonstrated the interaction of Par3 and Pk3 in vivo, extending our initial observations in cultured cells. This technique detects both stable and transient protein-protein interactions (*Roux et al., 2018*), and will help dissect cell signaling events leading to PCP. Proximity biotinylation allowed us to detect not only the novel interaction of Par3 and Pk3, but also the known association of Vangl2 and Pk3 (*Bastock et al., 2003*; *Chu et al., 2016*; *Chu and Sokol, 2016*; *Jenny et al., 2003*), confirming validity of this approach. Consistent with the critical role of the Par3/Pk3 interaction, we show that the Pk3-binding fragment Par3[272-544] specifically inhibits asymmetric distribution of PCP complexes in the neuroepithelial cells. Moreover, the Par3 construct that does not bind Pk3 failed to interfere with body axis elongation that is commonly associated with PCP signaling (*Gray et al., 2011*; *Sokol, 2000*; *Wallingford et al., 2002*).

The ability of Par3 to bind Pk3 and recruit it to the apical membrane may be a prerequisite for the formation of functional PCP complexes at the apical anterior cell borders (*Figure 7J*) (*Ossipova et al., 2015a*; *Ossipova et al., 2015c*). Consistent with this hypothesis, we found that in cultured cells Par3 associated with Vangl2 only when Pk3 was coexpressed, suggesting the formation of the ternary complex between Par3, Pk3 and Vangl2. Moreover, both the dominant interfering Par3N construct and Par3 depletion reduced the biotinylation of Vangl2 by BL-Pk3, supporting the view that Par3-dependent apical recruitment of Pk3 is essential for PCP complex formation. Whereas the direct binding of Pk3 by Par3 is the simplest interpretation of our data, alternatively, Par3 may recruit Pk3 to the apical domain by altering apical membrane properties (*Ahmed and Macara, 2017*; *Bryant et al., 2010*; *Ruch et al., 2017*). It is currently unknown whether the Par3/Pk3 interaction is modulated by other Par3-associated proteins, such as Par6, JAM-A or Nectin (*Ebnet et al., 2001*; *Joberty et al., 2000*; *Takekuni et al., 2003*).

Taken together, our experiments indicate that the interaction of Par3 with the core PCP machinery is required for the formation of apical PCP complexes. These findings parallel studies in *Drosophila* blastoderm where Par3 stimulates E-cadherin recruitment to apicolateral junctions (*McGill et al., 2009*). Par3 also promotes intestinal epithelial cell polarization by interacting with E-cadherin and other polarity proteins in *C. elegans* embryos (*Achilleos et al., 2010*). Apical enrichment of PCP complexes that we observe may be directly responsible for neural plate PCP or can modulate other cellular processes that ultimately contribute to neural tube closure, such as radial and mediolateral cell intercalation, apical constriction or centrosome and cilia functions. Of note, both Par3 (*Werner et al., 2014*) and core PCP proteins (*Ossipova et al., 2015a*) were implicated in radial intercalation of multiciliated epidermal cells. Moreover, both Par3 and Pk3 localize at the centrosome and may control centrosome organization and cilia growth (*Chu et al., 2016*; *Feldman and Priess, 2012*; *Inaba et al., 2015*; *Jakobsen et al., 2011*). Further experiments are necessary to address these possibilities.

# Materials and methods

## Key resources table

| Reagent type (species) or resource | Designation | Source or reference | Identifiers | Additional information |
|---|---|---|---|---|
| Gene (*Mus musculus*) | Par3 | PMID: 10934475 | accession number AY026057 | |
| Gene (*Xenopus laevis*) | Par3 | PMID: 25070955 | Xl.16888 | Brian Mitchell lab |
| Gene (*X. laevis*) | Prickle3 (Pk3) | PMID: 27062996 | GenBank BC154995 | Sergei Sokol lab |
| Gene (*X. laevis*) | Vangl2 | PMID: 27658614 | GeneID: 398271 | Sergei Sokol lab |

*Continued on next page*

*Continued*

| Reagent type (species) or resource | Designation | Source or reference | Identifiers | Additional information |
|---|---|---|---|---|
| Gene (*Aquifex aeolicus*) | Biotin Ligase (BL) | PMID: 26912792 | | Sergei Sokol lab |
| Genetic reagent (*X. laevis*) | Par3 morpholino, Par3MOATG | this paper | | |
| Genetic reagent (*X. laevis*) | Par3 morpholino, Par3MO5'UTR | this paper | | |
| Genetic reagent (*X. laevis*) | Vangl2 MO | PMID: 26079437 | | |
| Cell line (*Homo sapiens*) | HEK293T | ATCC | RRID:CVCL_0063 | |
| Antibody | anti-PKCζ (rabbit polyclonal, C20) | Santa Cruz | RRID:AB_2300359 | (1:200 IHC) |
| Antibody | anti-Biotin (goat polyclonal) | Cell Signaling | RRID:AB_10696897 | (1:3000 IB) |
| Antibody | anti-β-Catenin (rabbit polyclonal) | Sigma | RRID:AB_476831 | (1:200 IHC) |
| Antibody | anti-FLAG (mouse monoclonal, M2) | Sigma | RRID:AB_439685 | (1:1000 IB) |
| Antibody | anti-GFP (mouse monoclonal, B2) | Santa Cruz | RRID:AB_627695 | (1:100 IHC, 1:4000 IB) |
| Antibody | anti-GFP (rabbit polyclonal) | Invitrogen | RRID:AB_221569 | (1:600) |
| Antibody | anti-HA (mouse monoclonal, 12CA5) | NA | RRID:AB_2532070 | (1:100 IHC, 1:1000 IB) |
| Antibody | anti-HA (rabbit polyclonal) | Bethyl Laboratories | RRID:AB_67465 | (1:3000 IB) |
| Antibody | anti-Myc (mouse monoclonal, 9E10) | | RRID:CVCL_G671 | (1:60 IHC) |
| Antibody | anti-Par3 (rabbit polyclonal) | Millipore | RRID:AB_11213581 | (1:200 IHC, 1:4000 IB) |
| Antibody | anti-Vangl2 (rabbit polyclonal) | PMID: 25910938 | RRID:AB_2744499 | (1:100 IHC, 1:500 IB) |
| Antibody | anti-ZO1 (mouse monoclonal) | Invitrogen | RRID:AB_2533147 | (1:200 IHC) |
| antibody | anti-ZO1 (rabbit polyclonal) | Zymed | RRID:AB_138452 | (1:200 IHC) |
| Recombinant DNA reagent | BL-Pk3 | this paper | | (plasmid) |
| Recombinant DNA reagent | BL-Vangl2 | this paper | | (plasmid) |
| Recombinant DNA reagent | GFP-C1 | PMID: 22778024 | | (plasmid) |
| Recombinant DNA reagent | GFP-CAAX | PMID: 24818582 | | (plasmid) |
| Recombinant DNA reagent | GFP-Dvl2 | PMID: 15720724 | | (plasmid) |
| Recombinant DNA reagent | GFP-Par3 | PMID: 25070955 | | (plasmid) |
| Recombinant DNA reagent | GFP-Par3N | PMID: 25070955 | | (plasmid) |
| Recombinant DNA reagent | RFP-Par3N | PMID: 25070955 | | (plasmid) |

*Continued on next page*

*Continued*

| Reagent type (species) or resource | Designation | Source or reference | Identifiers | Additional information |
|---|---|---|---|---|
| Recombinant DNA reagent | mCherry | PMID: 19096028 | | (plasmid) |
| Recombinant DNA reagent | Myc-Par1T560A | PMID: 17993468 | | (plasmid) |
| Recombinant DNA reagent | Myc-Par3 | this paper | | (plasmid) |
| Recombinant DNA reagent | Myc-Par3[1-271] | this paper | | (plasmid) |
| Recombinant DNA reagent | Myc-Par3[272-544] | this paper | | (plasmid) |
| Recombinant DNA reagent | HA-Par3[272-544] | this paper | | (plasmid) |
| Recombinant DNA reagent | HA-Par3[545-756] | this paper | | (plasmid) |
| Recombinant DNA reagent | Myc-Par3[545-756] | this paper | | (plasmid) |
| Recombinant DNA reagent | Myc-Par3[757–1035] | this paper | | (plasmid) |
| Recombinant DNA reagent | Myc-Par3[934–1334] | this paper | | (plasmid) |
| Recombinant DNA reagent | Myc-Par3ΔΔ | this paper | | (plasmid) |
| Recombinant DNA reagent | HA-RFP-Pk3 | this paper | | (plasmid) |
| Recombinant DNA reagent | HA-Vangl2 | PMID: 27658614 | | (plasmid) |
| Recombinant DNA reagent | GFP-Pk3 | PMID: 27062996 | | (plasmid) |
| Recombinant DNA reagent | FLAG-Pk3 | PMID: 27062996 | | (plasmid) |
| Recombinant DNA reagent | FLAG-GFP-Pk3 | PMID: 27062996 | | (plasmid) |
| Recombinant DNA reagent | FLAG-Pk3ΔPET | PMID: 27062996 | | (plasmid) |
| Recombinant DNA reagent | FLAG-GFP-Pk3ΔPET | PMID: 27062996 | | (plasmid) |
| Recombinant DNA reagent | turboFP635 | this paper | | (plasmid) |
| Immuno precipitation reagent | Myc-Trap beads | Chromotek | | |

## Plasmids, mRNA synthesis and morpholino oligonucleotides (MOs)

Plasmids encoding FLAG and GFP-tagged *Xenopus* Pk3 and Pk3ΔPET and pCS2-HA-Vangl2 (*Chu et al., 2016*), GFP-C1 (*Kim et al., 2012*), Myc-Par1T560A (*Ossipova et al., 2007*), GFP-Dvl2 (*Itoh et al., 2005*), membrane-attached GFP-CAAX (*Ossipova et al., 2014*) and mCherry (*Choi and Sokol, 2009*) have been previously described. TurboFP635-pCS2 was made from the TurboFP635 (Katushka) plasmid obtained from A. Zaraisky. The cDNA insert encoding mouse Par3 (a gift of Tony Pawson, accession number AY026057) was subcloned into pCS2-Myc. GFP-tagged *Xenopus laevis* Par3, GFP-Par3N and RFP-Par3N in pCS2tub vector were gifts of Brian Mitchell (*Werner et al., 2014*). GFP- and RFP-Par3N constructs were subcloned into pCS2 + vector. HA-RFP-Pk3-pCS2 was generated by PCR. Fragments of mouse Par3 cDNA corresponding to amino acids 1–271; 272–544; 545–756; 757–1035; and 934–1334 were amplified by PCR and subcloned into pCS2-Myc or pCS2-

HA. In Par3ΔΔ, the DNA fragments corresponding to amino acids V274-V544 and G1036-S1334 were deleted. pCS2-FLAG-BL-Pk3 and pCS2-FLAG-BL-Vangl2 encode the amino acids N3-L185 of modified biotin ligase (BioID2) from *Aquifex aeolicus* (*Kim et al., 2016*) fused in-frame to the N-termini of *Xenopus laevis* Pk3 and Vangl2, respectively. Details of cloning are available upon request. All DNA constructs were verified by sequencing.

Capped mRNAs were synthesized using mMessage mMachine kit (Ambion, Austin, TX). For depletion studies, the following MOs were purchased from Gene Tools (Philomath, OR): Par3-MO$^{ATG}$: 5'-AGCTCACAGTCACCTTCATCCTGCG-3'; Par3MO$^{5'UTR}$: 5'- CAGGGTTCCCGTATTCCAC TCCGTG −3' control MO1 (CO MO1), 5'-GCTTCAGCTAGTGACACATGCAT-3'; control MO2 (CO MO2), 5'- AGCGTTTCAGGCCGATCTCTCAGTC-3'. Vangl2 MO 5'-GAGTACCGGCTTTTGTGGCGA TCCA-3' (*Ossipova et al., 2015a*).

## *Xenopus* embryo culture, microinjections, and phenotypic analysis

In vitro fertilized eggs were obtained from *Xenopus laevis* PRID:NXR_0.0095 and cultured in 0.1x Marc's Modified Ringer's solution (MMR) (*Newport and Kirschner, 1982*) as described previously (*Itoh et al., 2005*). Staging was according to (*Nieuwkoop and Faber, 1994*). For microinjection, four-to-16-cell embryos were transferred to 3% Ficoll in 0.5 x MMR (50 mM NaCl, 1 mM KCl, 1 mM CaCl$_2$, 0.5 mM MgCl$_2$, 2.5 mM HEPES, pH 7.4) and injected with 5–10 nl of a solution containing RNAs and/or MOs. For mosaic expression of PCP complexes, embryos were injected into two dorsal blastomeres of 16–32 cell embryos. Amounts of injected mRNAs per embryo have been optimized in preliminary dose-response experiments (data not shown) and are indicated in figure legends.

For phenotype analysis, frequencies of neural tube defects were calculated as means ± s. d. In unilaterally injected embryos, neural tube was scored as defective when the distance between the neural fold and the midline (white line) at the injected side was at least 1.5 times of that at the uninjected side. Body axis extension was estimated by measuring the length of stage 26 embryos. Blastopore defects were scored at stage 12, the defects were considered mild if the blastopore diameter was more than twice of that of uninjected embryos and severe when the blastopore groove was not visible.

## Cell culture and transfection

HEK293T cells (ATCC) were maintained in Dulbecco's modified Eagles medium (Corning) with 10% fetal bovine serum (Sigma) and penicillin/streptomycin (Sigma). This cell line was tested and found negative for mycoplasma contamination. Cells growing at 50–70% confluence were transiently transfected using linear polyethylenimine (M.W. 25000, Polysciences) as described (*Ossipova et al., 2009*). Each 60 mm dish of cells received 1 μg of pCS2 plasmids encoding FLAG-Pk3, FLAG-GFP-Pk3ΔPET and Myc-Par3 constructs. For transfections, pCS2 was added to plasmid DNA mixture to reach the total DNA amount of 3 μg.

## Immunostaining and fluorescent protein detection, imaging and quantification

For analysis of protein localization, embryos were collected at gastrula or early neurula stages and the vitelline membrane was removed manually. To quantify PCP in the neural plate, GFP-Pk3 RNA was co-expressed with mCherry and Vangl2 RNAs at the previously established doses that have no effect on normal development. Stage 14 embryos were fixed, neural plate explants were dissected and scored. For preparation of neural plate explants, embryos were fixed with 4% formaldehyde in phosphate buffered saline (PBS) for 40 min, washed in PBS and dissected. Cryosectioning and immunostaining were performed as described (*Dollar et al., 2005*). Embryos were fixed in 2% trichloroacetic acid for 30 min followed by the 30 min wash in 0.3% Triton X100 in 1x PBS for analysis of endogenous Par3 and Vangl2 as described (*Ossipova et al., 2015b*). For analysis of apicobasal distribution of GFP-Pk3, HA-RFP-Pk3, GFP-Dvl2, Myc-Par3, Myc-Par1T560A and endogenous Par3 proteins embryos at st. 10.5–11 were fixed overnight at 4°C in Dent's fixative (80% methanol, 20% dimethylsulfoxide). Antibodies against the following antigens were used: GFP (1:200, B-2, mouse monoclonal, Santa Cruz or rabbit polyclonal, Invitrogen), ZO-1 (1:200, mouse, Invitrogen; rabbit, Zymed), β-catenin (1:200, rabbit polyclonal, Sigma), anti-Myc (1:60, 9E10 hybridoma supernatant or rabbit monoclonal 1:600, Cell Signaling), anti-HA (1:100, 12CA5 hybridoma supernatant), anti-Par3

(1:4000, Millipore), anti-PKCζ(1:200, C20, rabbit polyclonal, Santa Cruz). Secondary antibodies were against mouse or rabbit IgG conjugated to Alexa Fluor 488, Alexa Fluor 555 (1:100, Invitrogen) or Cy3 (1:100, Jackson ImmunoResearch). Cryosections and explants were mounted for observation with the Vectashield mounting medium (Vector). Standard specificity controls were performed to confirm lack of cross-reactivity and no staining without primary antibodies. Images that are representative of at least 10 different fields were captured using a Zeiss AxioImager microscope with the Apotome attachment (Zeiss, Germany). The data shown are from two to five independent experiments with 5–15 embryos per group. Quantification for Par3 and ZO1 distribution in the neuroepithelium has been carried out using ImageJ as described (*McGreevy et al., 2015*). Significance was determined by two-tailed T test (Microsoft Excel).

## Immunoprecipitation and immunoblotting

For immunoprecipitation, cells transfected for 48 hr were lysed in the buffer (50 mM Tris-HCl at pH 7.5, 150 mM NaCl, 1 mM EDTA, 1% TritonX-100, 1 mM $Na_3VO_4$, 10 mM NaF), containing cOmplete Mini EDTA-free protease inhibitor cocktail (Roche). After centrifugation at 16,000 g, the supernatant was incubated with anti-FLAG agarose beads (Sigma) at 4°C for 2 hr or with anti-Myc antibodies (9E10) for 2 hr and Protein A Sepharose (GE Healthcare) at 4°C for 2 hr. Myc-trap beads (Chromotek) were used to pull-down Par3-containing protein complexes. The beads were washed three times in lysis buffer, and subjected to SDS-PAGE and immunoblotting using standard protocols (*Gloy et al., 2002*). Chemiluminescence was acquired by the ChemiDoc MP imager (BioRad) and band intensities were quantified by the accompanying software (BioRad).

## Proximity biotinylation in *Xenopus* embryos

For proximity biotinylation embryos were injected into the animal pole of four- to eight-cell embryos with 10 nl of the solution containing 1.6 mM of biotin and 0.1–0.5 ng of RNAs encoding FLAG-BL-Pk3, FLAG-BL-Vangl2 and Myc- or GFP-tagged Par3 (0.1 ng), or HA-Vangl2 (50 pg). Embryos were collected at stages 13–14 and protein biotinylation was assessed in embryo lysates and pulldowns with mouse anti-HA (12A5) and anti-Myc (9E10) hybridoma supernatants, rabbit anti-Par3 (07–330, Millipore) antibodies. Proteins were detected by immunoblotting with goat anti-biotin-HRP antibodies (Cell Signaling), rabbit anti-Par3 and anti-HA (A190-108A, Bethyl Labs), and mouse anti-Myc, anti-FLAG (M2 Sigma) and anti-GFP (sc9996, Santa Cruz) antibodies.

## Acknowledgement

We thank Tony Pawson and Brian Mitchell for plasmids, and Bo Xiang for advice on biotinylation assays in *Xenopus*. We are grateful to Florence Marlow and Chih-Wen Chu for comments on the manuscript and members of the Sokol laboratory for discussions. This study was supported by NIH grants GM122492 and NS100759 to S Y S.

## Additional information

### Funding

| Funder | Grant reference number | Author |
| --- | --- | --- |
| National Institutes of Health | GM122492 | Sergei Y Sokol |
| National Institutes of Health | NS100759 | Sergei Y Sokol |

The funders had no role in study design, data collection and interpretation, or the decision to submit the work for publication.

### Author contributions

Ilya Chuykin, Conceptualization, Validation, Investigation, Visualization, Methodology, Writing—original draft; Olga Ossipova, Investigation, Visualization, Writing—original draft; Sergei Y Sokol, Conceptualization, Supervision, Funding acquisition, Methodology, Writing—review and editing

## Author ORCIDs

Ilya Chuykin (iD) http://orcid.org/0000-0002-1773-6829
Sergei Y Sokol (iD) http://orcid.org/0000-0002-3963-9202

## Ethics

Animal experimentation: This study was carried out in strict accordance with the recommendations in the Guide for the Care and Use of Laboratory Animals of the National Institutes of Health. The protocol 04-1295 was approved by the IACUC of the Icahn School of Medicine at Mount Sinai.

## Decision letter and Author response

Decision letter https://doi.org/10.7554/eLife.37881.021
Author response https://doi.org/10.7554/eLife.37881.022

## Additional files

### Supplementary files

• Transparent reporting form
DOI: https://doi.org/10.7554/eLife.37881.019

### Data availability

All data generated or analyzed during this study are included in the manuscript and supporting files.

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
