## [Decision Letter]

Thank you for submitting your article "Par3 interacts with Prickle3 to generate apical planar cell polarity complexes in the vertebrate neural plate" for consideration by *eLife*. Your article has been reviewed by three peer reviewers, including Yukiko M Yamashita as the Reviewing Editor and Reviewer #1, and the evaluation has been overseen by Marianne Bronner as the Senior Editor.

The reviewers have discussed the reviews with one another and the Reviewing Editor has drafted this decision to help you prepare a revised submission.

The authors of this manuscript reveal the role of Par3 apical-basal polarity protein in regulation of planar cell polarity. They found planar polarization of Par3 in the *Xenopus* neural plate. Morpholino-based Par3 knockdown revealed defects in polarization of planar cell polarity proteins Vangl2 and Pk3. The authors used co-IP and proximity biotinylation assays to show binding of Par3 with Pk3 and identify the critical domains in Par3 for the binding, and reveal an important role of Par3 in localization of Pk3 to the apical membrane domain. They show that the overexpression of a Prickle3-binding Par3 fragment disrupted PCP in the neural plate. The authors finally demonstrate that Par3 is necessary to promote Pk3-Vangl2 interaction.

This is an exciting study that reveals novel role for the vertebrate Par3 protein and discovers a novel physical and functional connection between apical-basal and planar cell polarity pathways. The experiments are well done and the interpretations are appropriate. Conclusions are well supported by the evidence. The reviewers raised the following major concerns to be addressed prior to publication.

Essential revisions:

1) The authors assume that Par3 is upstream of Pk3/Vangl2. What if Par3 and Pk3/Vangl2 reinforce each other localization and function? Vangl2 is not showing proper planar polarization without Par3 (Figure 2). Is Par3 showing planar polarization without Vangl2?

2) The authors assume that expression of Par3[274-544] generated phenotypes because it disrupts interaction between endogenous Par3 and Pk3. This disruption of Par3 and Pk3 interaction should be confirmed directly using conventional co-IP approach in cell line in culture.

3) How Par3 promotes Pk3-Vangl2 interaction? What may be the molecular mechanism responsible? Is Par3 promoting interaction between Pk3 and Vangl2 in cell line overexpression assays?

4) Why GFP-Pk3 is not localizing to the apical domain at all without Par3 overexpression (Figure 6D)? The endogenous Par3 should be at the apical membrane domain and it should direct at least some GFP-Pk3 to the apical membrane domain.

5) The authors show that Par3 depletion does not affect apicobasal polarity. Is this due to low doses of the MO used, or is it because of a distinct function of Par3 in *Xenopus* ectoderm? At higher doses, does Par3 MO interfere with apicobasal polarity?

6) In the presence of Par3 MO, Vangl2 is no longer seen at the apical surface. Is the protein now localized to the basolateral membrane or is it internalized into membrane vesicles?

7) The effect of Par3 MO on Vangl2 distribution seems to differ from that of Par3 domains or deletion mutants on Pk3 distribution. To conclude that Par3 interacts with Pk3/Vangl2 complex, the experiments should be performed under similar conditions. For example, what happens to Pk3 with Par3 MO, or what is the distribution pattern of Vangl2 when Par3[272-544] construct is used?

8) Does Par3 compete with Vangl2 for Pk3, or they form a ternary complex?

9) The effect of Par3-N and Par3[272-544] on distribution of wild type Par3 should be shown, so that the mislocalization of Pk3 and the localization of endogenous Par3 can be compared and correlated.

---

## [Author Response]

Essential revisions:1) The authors assume that Par3 is upstream of Pk3/Vangl2. What if Par3 and Pk3/Vangl2 reinforce each other localization and function? Vangl2 is not showing proper planar polarization without Par3 (Figure 2). Is Par3 showing planar polarization without Vangl2?

We agree that Par3 and Vangl2 may potentially exhibit feedback regulation during the establishment of neural plate PCP. We performed a complementary experiment suggested by reviewers and analyzed Par3 localization in neural plates of Vangl2-depleted embryos. Indeed, Par3 planar polarity has been compromised in neuroepithelial cells depleted of Vangl2, indicating that planar polarization of Par3 requires core PCP proteins (new Figure 2—figure supplement 4, and subsection “Par3 plays an essential role in neural plate PCP”).

2) The authors assume that expression of Par3[274-544] generated phenotypes because it disrupts interaction between endogenous Par3 and Pk3. This disruption of Par3 and Pk3 interaction should be confirmed directly using conventional co-IP approach in cell line in culture.

We found that, in HEK293T cells, the levels of Pk3 and Pk3∆PET were upregulated by Par3[272-544], an effect that we did not observe in vivo. We, therefore, examined how Par3[272-544] influences the Prickle3-Par3 association directly in *Xenopus* ectoderm. The interaction was measured in a proximity biotinylation assay, as the degree of biotinylation of Par3 in the presence of Biotin Ligase-tagged Prickle3. Par3[272-544] had a negative effect on this protein interaction (new Figure 5—figure supplement 1),supporting the view that it competes with Par3 for Prickle3 binding. Moreover, Par3[272-544] was itself biotinylated, confirming its association with Prickle3. This new experiment is described in the first paragraph of the subsection “Functional significance of the Par3-Pk3 interaction for PCP”.

3) How Par3 promotes Pk3-Vangl2 interaction? What may be the molecular mechanism responsible? Is Par3 promoting interaction between Pk3 and Vangl2 in cell line overexpression assays?

We have carried out several experiments addressing the mechanism of Par3 effects on PCP. We did not see the competition of Par3 with Vangl2 for Pk3 binding, on the contrary, we show that Vangl2 binds to Par3 in the presence of Prickle3 (new Figure 4E). Vangl2 was not detected in pulldowns with Par3 mutant lacking Pk3 binding domains, Par3∆∆, indicating that Pk3 is necessary for the Par3-Vangl2 association (new Figure 4E). The simplest interpretation of this result is the formation of the ternary complex between Par3, Prickle3 and Vangl2. Since the promoting effect of Par3 on the total amount of the Pk3-Vangl2 complex was not apparent (Figure 4E), we propose that Par3 stimulates the interaction of Prickle3 and Vangl3 within the ternary complex.

We describe this experiment in the first paragraph of the subsection “The association of Par3 with the Pk3/Vangl2 complex and the identification of Pk3-interacting domains” and in the fourth paragraph of the Discussion.

4) Why GFP-Pk3 is not localizing to the apical domain at all without Par3 overexpression (Figure 6D)? The endogenous Par3 should be at the apical membrane domain and it should direct at least some GFP-Pk3 to the apical membrane domain.

We suspect that endogenous Par3 is fully engaged in preexisting binding complexes, including those containing different Prickle family members. We suspect that it may be simply unavailable for the recruitment of exogenous Prickle3.

5) The authors show that Par3 depletion does not affect apicobasal polarity. Is this due to low doses of the MO used, or is it because of a distinct function of Par3 in Xenopus ectoderm? At higher doses, does Par3 MO interfere with apicobasal polarity?

We have examined several apical and basal polarity markers at different doses of the MO and at additional developmental stages. These new experiments confirm our initial conclusions. Although the knockdown of Par3 is efficient and a strong effect on Vangl2 localization has been observed (new Figure 2—figure supplement 2), we saw no obvious changes in ZO1, aPKC or β-catenin (new Figure 2—figure supplement 3).

Similarly, no defects in apicobasal polarity were reported in *Drosophila* follicular epithelial cells in Bazooka/Par3 mutants (Shahab et al., 2015) and in mouse retinal endothelial cells with conditionally knocked out *par3* (Hikita et al., 2018). Together, these findings suggest that the requirement for Par3 in apicobasal polarity may depend on the cell type and developmental context.

6) In the presence of Par3 MO, Vangl2 is no longer seen at the apical surface. Is the protein now localized to the basolateral membrane or is it internalized into membrane vesicles?

Cryosections revealed that Vangl2 is enriched basolaterally rather than apically after Par3 depletion (Figure 2—figure supplement 2).This was described in the subsection “Par3 plays an essential role in neural plate PCP”.

7) The effect of Par3 MO on Vangl2 distribution seems to differ from that of Par3 domains or deletion mutants on Pk3 distribution. To conclude that Par3 interacts with Pk3/Vangl2 complex, the experiments should be performed under similar conditions. For example, what happens to Pk3 with Par3 MO, or what is the distribution pattern of Vangl2 when Par3[272-544] construct is used?

We found that the staining with our Vangl2 antibodies produces a strong background in cells expressing Par3[272-544] construct. However, a new experiment with Par3 MO demonstrated that Pk3/Vangl2 patches are not anteriorly polarized in Par3-depleted cells (Figure 7—figure supplement 2). This result is added to the second paragraph of the subsection “Par3 is required for the formation of the apical PCP complex in the neural plate”.

8) Does Par3 compete with Vangl2 for Pk3, or they form a ternary complex?

Please see the response to point 3.

9) The effect of Par3-N and Par3[272-544] on distribution of wild type Par3 should be shown, so that the mislocalization of Pk3 and the localization of endogenous Par3 can be compared and correlated.

Prompted by the reviewers, we examined the localization of myc-Par3 in cells expressing Par3N and Par3[272-544]. Par3[272-544] did not affect apical localization of Par3 (Figure 5—figure supplement 2) as described in the first paragraph of the subsection “Functional significance of the Par3-Pk3 interaction for PCP”, consistent with our assumption that the primary effect of this construct is on Pk3 binding. Par3N caused a pronounced decrease in apical Par3, as shown previously in cultured cells (Mizuno et al., 2003). This result is shown in Figure 7—figure supplement 1 and mentioned in the first paragraph of the subsection “Par3 is required for the formation of the apical PCP complex in the neural plate”.